# Innovation in Non-Invasive Diagnosis and Disease Monitoring for Meningiomas

**DOI:** 10.3390/ijms25084195

**Published:** 2024-04-10

**Authors:** Brianna Korte, Dimitrios Mathios

**Affiliations:** Department of Neurosurgery, Washington University Medical Campus, St. Louis, MO 63110, USA

**Keywords:** meningioma, brain tumor, liquid biopsy, biomarker, circulating tumor DNA, extracellular vesicle, micro-RNA, spectroscopy, seroreactivity

## Abstract

Meningiomas are tumors of the central nervous system that vary in their presentation, ranging from benign and slow-growing to highly aggressive. The standard method for diagnosing and classifying meningiomas involves invasive surgery and can fail to provide accurate prognostic information. Liquid biopsy methods, which exploit circulating tumor biomarkers such as DNA, extracellular vesicles, micro-RNA, proteins, and more, offer a non-invasive and dynamic approach for tumor classification, prognostication, and evaluating treatment response. Currently, a clinically approved liquid biopsy test for meningiomas does not exist. This review provides a discussion of current research and the challenges of implementing liquid biopsy techniques for advancing meningioma patient care.

## 1. Introduction

Meningiomas are frequently encountered brain tumors, comprising approximately 39.7% of all central nervous system (CNS) tumors and 55.4% of all non-malignant CNS tumors [1]. Meningioma tumors originate from meningothelial arachnoid cap cells within the brain or spinal cord [2].

The World Health Organization (WHO) classifies meningiomas into three grades based on their histological features: grade I (benign), grade II (atypical), and grade III (anaplastic) [3], with the majority presenting as benign (80.1%) [1]. Considering their classifications, the prognosis for meningiomas remains relatively favorable, with the 5-year progression-free survival being 95.7%, 81.8%, and 46.7% for grades I, II, and III, respectively [4]. However, meningioma patient outcomes can be variable and are not reliably predicted using traditional WHO grading, particularly among WHO grade I and II meningiomas [5]. A new meningioma grading algorithm combines multiple classification approaches including copy number alterations, mutations, methylation patterns, and WHO grade to return an integrated risk score that predicts recurrence risk more accurately and offers a precise risk assessment [5]. This score classifies meningiomas into three risk categories: low, intermediate, and high.

Low-risk meningiomas are often characterized by their slow growth and prolonged latency period of 20–30 years [6,7]. This slow progression can lead patients to present as asymptomatic or be diagnosed incidentally. As a result, patients are frequently placed under observation without immediate treatment, contingent on their symptoms and the tumor growth rate [8,9]. Conversely, high-risk meningiomas can be aggressive and require immediate attention. Therefore, this group of tumors can be quite heterogeneous in their natural history, which can make their initial management challenging. This presents an unmet clinical need for alternative ways other than traditional imaging to predict tumor aggressiveness, clinical course, and treatment response.

Traditional diagnostic and monitoring approaches for meningiomas rely on magnetic resonance imaging (MRI), computer tomography (CT), and tissue biopsy [2,8], but these methods have several limitations. Imaging, such as MRI, can occasionally misinterpret other intracranial lesions as meningiomas [8,10,11]. Dural-based tumors that mimic meningiomas on MRI include solid cancers or hematopoietic malignancies [12]. In cases such as those mentioned above, it is important to have a rapid diagnostic method to distinguish them from meningiomas, as a “watch and wait” strategy in malignant tumors is harmful for the patients. In addition, traditional imaging lacks sensitivity or specificity for detecting small tumors or residual disease [13] and provides limited prognostic information [13,14].

When treatment is opted for, it typically involves surgical resection, followed by radiation as necessary [15]. Prior to treatment, pathological assessment and molecular diagnosis are essential to guide these decisions, with tissue biopsies offering reliable histopathologic analysis, subtyping, and grading [8]. There are many obvious disadvantages to intracranial biopsies, as they carry inherent risk, contribute to patient anxiety, and are occasionally infeasible due to their location near essential brain regions [10]. Additionally, they may not fully capture tumor heterogeneity [16], and repeated biopsies for tumor monitoring are challenging. Given these limitations and the demand for non-invasive diagnostic or surveillance methods for CNS tumors, liquid biopsy techniques have gained prominence as potential non-invasive solutions.

Liquid biopsy involves analyzing circulating tumor biomarkers in bodily fluids, such as blood (plasma and serum), cerebrospinal fluid (CSF), urine, and saliva, as a minimally invasive approach for monitoring disease [17,18]. Common tumor biomarkers that can be exploited for liquid biopsy include circulating tumor cells (CTCs), circulating tumor DNA (ctDNA), microRNA (miRNA), extracellular vesicles (EVs), proteins, epigenetic signatures, and more. In recent years, there have been major advances in utilizing liquid biopsy techniques to detect, characterize, and prognosticate various cancers, such as lung [19], breast [20], and colorectal malignancies [21]. Liquid biopsy can supplement traditional imaging and biopsy methods to provide more robust and detailed diagnostics (Figure 1).

Unfortunately, applying liquid biopsy methods to intracranial tumors has posed unique obstacles, primarily due to a scarcity of measurable tumor biomarkers. One prevailing hypothesis suggests that the blood-brain barrier may hinder the shedding of tumor cells and biomarkers into bodily fluids [16,22,23,24], which contributes to the relative lack of success in intracranial tumor liquid biopsy research. Furthermore, biomarkers shed by other non-cancer tissues add background noise and further dilute the tumor-specific signals. Despite these obstacles, the exploration of liquid biopsy methods across all clinical stages of intracranial tumors, from early identification to surveying recurrence, remains highly significant and timely.

There are currently no clinically validated liquid biopsy prognostic or diagnostic biomarkers specific for meningiomas. As meningiomas can vary significantly in their natural history and biological behavior and the decision to treat these tumors early on at their presentation is crucial to maximize oncological control and quality of life, an accurate non-invasive biomarker of disease activity could provide an avenue for improved risk stratification and outcomes. Specifically, clinical uses for liquid biopsy methods would allow for distinction between meningiomas and brain metastases, discriminating aggressive from benign meningiomas, and monitoring disease progression. This review examines the current state of research on meningioma liquid biopsy, including a range of various biomarkers that have been studied and the ongoing challenges to be addressed.

## 2. Methods

Five markers or liquid biopsy approaches, including “cfDNA”, “miRNA”, “EV’s”, “seroreactivity”, and “spectroscopy”, were chosen for analysis based on reviews and research papers about liquid biopsy in brain tumors [16,17,18,25,26,27,28]. The marker or liquid biopsy approach was searched in Google Scholar and PubMed according to the search strategies outlined in Appendix A between 21 September 2023 and 20 March 2024. A total of 2040 articles were initially collected; 1419 articles underwent screening, and 25 were selected for analysis in this review (Figure 2) [29]. Eligible studies were selected based on the following criteria:The studies were full essays written in English.The patient cohort contained at least five meningioma patients.The study used one of the five liquid biopsy methods.At least one significant biomarker was observed.

## 3. Circulating Tumor DNA

Cell-free DNA (cfDNA) has proved to be a promising biomarker across several cancer types, including liver [30], lung [31], and breast cancer [32]. cfDNA represents the fraction of DNA circulating within biofluids upon apoptosis, necrosis, or active secretion and is primarily nucleosomal [33]. While the majority of cfDNA originates from natural white blood cell turnover [33,34], in individuals with cancer, cfDNA is also comprised of circulating tumor DNA (ctDNA) that has been released by tumor cells. Tumor-specific characteristics of ctDNA such as mutations, methylation, fragmentation patterns, nucleosome positioning, and histone modifications offer valuable insights into disease activity and aggressiveness and are sensitive enough to detect minimal residual disease (MRD) after curative-intent treatment [35,36,37,38]. Moreover, the short 2-h half-life of plasma cfDNA, coupled with its capacity to capture tumor heterogeneity, positions it as a reliable and dynamic marker for assessing tumor progression over time. The challenge with ctDNA as a biomarker arises from its low abundance in circulation, which is thought to constitute a fraction ranging from 0.01% to 10% of all cfDNA [39,40], although ctDNA concentration is variable among patients and its biological signal can be diluted further with the release of high quantities of non-tumor cfDNA dependent on clinical context.

Isolating ctDNA in patients with brain tumors presents a distinctive challenge, given that the abundance of ctDNA is especially scarce. In one pan-cancer study, only 10% of patients with gliomas exhibited detectable ctDNA in plasma, in stark contrast to the 82% detection rate among patients with non-brain tumors [39]. The low concentration of circulating ctDNA is thought to be related to tumor burden, location, and metabolism [33], the permeability of the blood-brain barrier [24], and the release of non-tumor DNA from immune cell death [33,34]. Additionally, the type of biofluid used for analysis impacts sensitivity, with CSF presenting higher concentrations of ctDNA on average compared to plasma in patients with brain tumors [41,42].

Given the numerous difficulties associated with discerning ctDNA in brain tumors, limited studies have examined its efficacy, with most of the research concentrating on glioblastoma. Nevertheless, cfDNA remains an advantageous biomarker for meningiomas for its various genetic and epigenetic attributes that can be analyzed.

### 3.1. ctDNA Mutations

Mutations found in tumor tissue biopsies are also reflected in ctDNA [43]. Therefore, an obvious application of liquid biopsy is genomic profiling for the presence of tumor-specific aberrations, including single nucleotide polymorphisms (SNPs), insertions, or deletions. Piccioni et al. [44] aimed to discern cfDNA genomic changes, specifically SNPs and gene amplifications, in a variety of patients with primary brain tumors, including meningiomas. A recognizable somatic alteration occurred in half of the cohort and 59% of the meningioma patients upon gene panel analysis. However, the study lacked concordant analysis with tumor tissue genetic analysis and the exclusion of mutations related to clonal hematopoiesis of indeterminate potential (CHIP).

Graillon et al. [45] endeavored to bridge the gap between tumor and liquid biopsy mutational analysis by validating mutations present in plasma or CSF liquid biopsy with known mutations in tumor tissue in 18 meningioma patients. However, a liquid biopsy of CSF or plasma did not indicate any mutations in 11 benign cases, and only confirmed three of seven mutations in aggressive cases [45]. These results are poor but suggest greater ctDNA circulation in more aggressive meningiomas. Confirmation of these results in a larger study is needed.

While these studies are an improvement compared to earlier findings [39], the approach of genotyping a limited number of genes for non-invasive diagnosis and molecular characterization of meningiomas provides limited sensitivity. Given the vast number of genomic and epigenomic aberrations that can be present in each tumor, the assay must be inclusive of a wide panel of possible tumor-specific molecular alterations to be sensitive in the setting of low amounts of ctDNA. Thus, liquid biopsy techniques for examining ctDNA mutations, although with obvious translational value, will have a more limited scope than anticipated in the landscape of liquid biopsy technology development.

### 3.2. Epigenomic Signatures

A liquid biopsy platform that can more thoroughly analyze the patterns and origin of cfDNA in circulation is cfDNA methylation. DNA methylation profiling in meningiomas has high clinical utility, as it has recently been exemplified to molecularly classify and risk stratify meningiomas [46,47]. Assessing meningioma methylation markers can predict recurrence risk with more precision than traditional WHO-grade risk [48]. In addition, DNA methylation plays a crucial role in regulating gene expression and chromosomal stability; thus, changes in methylation can also contribute to carcinogenesis [49]. For example, *TRAF7* and *KLF4* promoter methylation in meningiomas is linked to a higher tumor grade and risk of recurrence [50].

Methylation signatures in cfDNA provide genome-wide epigenetic information that can be used to discern cancer-associated immune responses and are informative regarding gene expression and meningioma classification. Nassiri et al. [51] successfully utilized cfDNA methylation profiles from plasma to differentiate various intracranial tumors with high accuracy. Upon establishment of the top 300 differently methylated regions in meningioma versus five other brain tumor types, a classification machine learning model was developed in which the area under the curve (AUC) was 0.89 [51].

Similarly, Herrgott et al. [52] identified congruent and specific methylation probes in meningioma tissue, serum, and plasma that could distinguish meningioma from other CNS tumors. Liquid biopsy diagnostic and prognostic models were subsequently developed using machine learning approaches that could predict the presence of meningioma and the risk of recurrence with an accuracy of 85% and 87.7%, respectively. Despite the limited number of patients and the retrospective nature of these studies, these results are optimistic, and enhancing the sensitivity and specificity of these models could pave the way for clinical implementation.

### 3.3. Multi-Omic Approaches

In addition to the various cfDNA characteristics discussed here, many other attributes of cfDNA can be exploited, including fragmentation patterns and nucleosome positioning. Yet, few have been studied in meningiomas.

Multi-omic feature analysis has recently gained prominence for its ability to evaluate several characteristics of cfDNA at once [53]. This strategy can combine multiple fields of study, including genomics, methylomics, fragmentomics, and nucleosomics, to provide more complete prediction models that can mitigate low ctDNA concentrations. Multi-modal feature selection combined with dimensionality reduction and machine learning can improve the sensitivity and specificity issues of single-omic approaches while mitigating the risk of overfitting. While this methodology has yet to be applied to meningiomas, multi-omic cfDNA strategies appear to be the next frontier in liquid biopsy research.

## 4. Extracellular Vesicles

Extracellular vesicles (EVs) engage in cell-to-cell communication and contain proteins, DNA, RNA, and lipids. EVs are secreted by normal and cancer cells into the bloodstream, and, due to the protective nature of the EV membrane, the contained biomolecules are promising tumor biomarkers. In addition, there is evidence that suggests EVs can pass through the blood-brain barrier via transcytosis [54], making them an excellent marker for CNS tumors. However, a significant challenge arises from the fact that tumor-derived EVs cannot be differentiated in the bloodstream from EVs originating from other cell types. Recent work has identified meningioma-specific EV proteins that have potential for distinguishing meningioma-originating EVs in circulation [55,56,57]. Nevertheless, there are currently no clinically applicable methods for using EVs in biofluids for discerning CNS tumors.

Due to their diversity of contained biological molecules, many components of EVs can be used for analysis. For example, Negroni et al. [58] and Abdelrahman et al. [59] examined the miRNA molecule miR-497 in circulating exosomes for its differential expression in meningioma patients and among meningioma WHO grades. In addition to discriminating between meningioma and healthy patients [59], miR-497 expression was shown to be significantly downregulated in WHO grade III meningiomas versus grades I and II, as shown by a receiver operating characteristic analysis (ROC) (AUC = 0.894) [58]. Another group of researchers, Ricklefs et al. [55], explored EV-DNA methylation patterns, EV proteomics, and EV-DNA mutations specific to meningioma tissue that could be applied to liquid biopsy. It was revealed that EV-DNA secreted by meningioma cells can provide successful tumor classification by reflecting tumor-specific methylation profiles, mutations, and copy number variations. The EV proteome, on the other hand, did not allow for successful tumor classification but has potential for probing meningioma-originating EVs. Additionally, it was noted that the concentration of EVs in plasma is elevated in patients with meningioma, and this increase correlates with the malignancy grade and extent of edema, a pattern also consistent in studies involving glioblastoma patients [60]. As these findings were characterized in meningioma tumor tissue, not through a liquid biopsy, the next step is to assess whether these findings are consistent with meningioma EV isolation from biofluids.

It has been shown that meningioma tumors exhibit increased expression of proteins such as EGFR, NEK9, EPS812, CKAP4, SET, and STAT2 [56]. Differential protein expression of meningioma-originating EVs, therefore, has the potential to distinguish meningiomas. Dobra et al. [61] (2020) aimed to establish a unique protein fingerprint of CNS tumors, including meningiomas, from serum-derived sEVs (small extracellular vesicles) [61]. Liquid chromatography and mass spectrometry were performed on pooled serum samples from four patient groups, including glioblastoma, benign meningioma, brain metastasis from non-small-cell lung cancer, and non-cancer controls. Upon further statistical analysis, Dobra et al. [61] (2020) successfully determined 17 sEV proteins that, when considered together, could discriminate between the patient groups. [61]. Liquid chromatography and mass spectrometry were performed on pooled serum samples from four patient groups, including glioblastoma, benign meningioma, brain metastasis from non-small-cell lung cancer, and non-cancer controls. Upon further statistical analysis, Dobra et al. [61] (2020) successfully determined 17 sEV proteins that, when considered together, could discriminate between the patient groups.

Dobra et al. [57] (2023) furthered their analysis and attempted to uncover a single sEV protein marker from the established group of 17 that could be indicative of prognosis. The matrix metalloproteinase-9 (MMP-9) protein was theorized as a potential single-protein marker, as it is known to have functional roles in cancer cell survival and metastases [62,63]. Upon analyzing blood serum from four patient groups, including meningioma, the group confirmed increased sEV MMP-9 concentration is associated with decreased patient survival. It was validated that MMP-9 concentration in circulation is related to tumor aggressiveness and showed variation among tumor type and stage; however, it was not successful in distinguishing between control and meningioma patient groups [57].

While these findings are promising, the use of EVs as liquid biopsy biomarkers is a new technique that has not been validated thoroughly. The limited number of studies in this area and the ongoing dilemma of delineating tumor originating EVs in biofluids are significant obstacles that need to be addressed.

## 5. miRNA

MicroRNAs (miRNA) participate in controlling the stability and translation of mRNA [22,64]. As such, miRNAs are involved in tumor development by acting as oncogenes or tumor suppressers [65,66,67,68] and miRNA expression profiles can distinguish human cancers [66]. In biofluids, miRNAs can be encapsulated in EVs or presented as free miRNAs.

Several studies have explored their role in meningiomas and have attempted to reveal specific miRNAs that are up- or downregulated in the blood of meningioma patients with real-time quantitative PCR. For example, Zhi et al. [69] uncovered six serum miRNAs with two-fold differences in expression in meningioma patients versus healthy controls and constructed a diagnostic model. Similarly, Kopkova et al. [70] observed nine differentially expressed miRNAs in CSF across various CNS tumors. Abdelrahman et al. [59] and Carneiro et al. [71] found that differential expression of even one to two miRNA molecules in plasma or serum has the potential to differentiate meningiomas and distinguish low- and high-grade meningiomas. However, utilizing only one or two miRNA biomarkers is unlikely to have practical use in the clinic, and these studies face many limitations, as the detected miRNAs are not necessarily tumor-derived, and the specificity and sensitivity of the models are low in comparison to other liquid biopsy approaches.

As up to 20% of surgically resected meningiomas recur [47], another application of utilizing miRNAs in liquid biopsy is to assess prognosis and recurrence risk. Urbschat et al. [72] examined the expression of four miRNAs in solid tumors and blood plasma in meningioma patients as potential prognostic markers. Their results suggest that miR-200a-3p could be a useful marker to predict meningioma recurrence, as its expression was significantly downregulated in the tumor tissue of recurrent meningiomas compared to new diagnoses [72]. However, the application of miR-200a-3p expression for liquid biopsy is limited since this differential expression between recurrent and non-recurrent meningioma tissue is discordant in plasma.

Another group of researchers developed a nanowire-based device that could be used to extract EV-encapsulated and free miRNAs from urine samples [73]. Upon microarray analysis, 23 differentially expressed miRNAs were discovered to characterize many CNS cancers, including meningioma, against individuals without cancer [73]. While this liquid biopsy technique is not meningioma-specific, the nanowire device has clinical potential for broad CNS cancer discovery. To evaluate robustness, continued research should assess this method’s ability to distinguish CNS cancers from other cancer types and apply this technique to larger patient cohorts.

Like other biomarkers, miRNAs have advantages and disadvantages for liquid biopsy applications. Another factor that complicates the use of miRNA expression as a biomarker of disease activity is establishing thresholds of differential expression above which differences are considered biologically relevant. In some cases, changes between 10 and 20-fold are considered relevant, while in others, a less than two-fold change has significant biological effects [66]. Altogether, there are many hurdles to overcome before miRNA can be considered a reliable, specific, and applicable biomarker.

## 6. Seroreactivity to Autoantibodies

Neoplasias such as meningiomas express tumor-associated antigens (TAAs), which can trigger the patients’ immune system to produce autoantibodies in defense [74,75]. In addition to targeting TAAs, these autoantibodies can be found in blood serum, which makes them a promising liquid biopsy biomarker. Following immunogenic antigen identification methods such as serologic identification of recombinantly expressed clones (SEREX), meningioma and control sera can be screened against antigen sets for reactive antibodies [75,76]. A panel of meningioma-associated antigens exhibiting reactivity specific to meningioma patient serum can then be applied as a non-invasive diagnostic tool.

Comtesse et al. [76] were the first researchers to analyze multi-antigen panels as a potential diagnostic method for meningiomas. They established a panel of 62 meningioma-associated antigens and evaluated the seroreactivity of meningioma and healthy sera. Their results showed that twice as many antigens were reactive to meningioma sera compared to healthy sera on average, and 17 antigens exhibited seroreactivity exclusive to meningioma, suggesting that an antigen panel could be used for distinguishing meningiomas.

Keller et al. [77] endeavored to produce a diagnostic model for meningiomas by applying statistical learning methods to the seroreactivity patterns of meningiomas and healthy sera. They used a 57 meningioma-associated antigen panel and returned a model with a sensitivity and specificity of 84.5% and 96.2%, respectively. In continuation of these results, Ludwig et al. [78] sought to determine if increasing the number of antigens could further improve classification. Seroreactivity was measured against a larger array of nearly two thousand antigen-expressing clones, and sensitivity and specificity were minimally improved to 91.8% and 95.6%, respectively.

Thioredoxin domain containing 16 (TXNDC16) is a known meningioma-associated antigen. In previous studies, seroreactivity has been exclusively reported in the sera of meningioma patients [79]. As such, Harz et al. [79] evaluated the ability of TXNDC16 antigens as meningioma specific biomarkers by assessing 163 TXDNC16 peptide arrays to identify specific TXNDC16 epitopes exclusively recognized by meningioma autoantibodies. While a single epitope was not recognized by all meningioma serum samples, a set of five immunogenic epitopes as determined by feature selection could successfully discriminate between meningioma and healthy serum with a sensitivity and specificity of 90% and 83.7% [79].

These results are promising; however, there are several disadvantages to using autoantibodies as a liquid biopsy marker. Many TAAs can be immunogenic across several cancer types, resulting in the same autoantibodies being produced in response to different neoplasias and limiting the specificity for differentiating cancer types [75]. In addition, TAAs can be immunogenic in other diseases, such as autoimmune conditions, and even be present in healthy individuals [75]. Therefore, to develop a robust diagnostic or prognostic method with high sensitivity and specificity, multi-antigen panels are required, as single antigen markers are limited.

## 7. Multi-Marker Spectroscopic Approach

Spectroscopic liquid biopsy techniques are notable for their breadth, encompassing a wide range of biological features and overcoming the limitations of single-marker methodologies. Instead of analyzing a single biomarker, a machine learning algorithm can interpret the complete biochemical spectra for classification. By default, spectroscopic liquid biopsy does not provide tumor genetic information and, therefore, cannot guide treatment decisions in its present application.

Attenuated total reflectance-Fourier transform infrared (ATR-FTIR) spectroscopy directs infrared light through biofluids, such as blood serum, to generate a specific biochemical spectrum, or fingerprint, which trained machine learning algorithms can discern to predict patients’ disease status [80,81]. ATR-FTIR has been investigated in colorectal [82], lung [83], ovarian [84], and breast cancer [85], and recently, has been applied to a variety of brain tumors, producing results with high sensitivity and specificity [86,87].

One of the early ATR-FTIR studies exemplified that such a technique could be applied for diagnostic purposes at a variety of levels, including cancer versus non-cancer, metastatic versus primary tumors, glioma versus meningioma, severity of the tumor, and more [88]. Since then, improvements in computational strategies, machine learning, and cheaper instrumentations have greatly increased the capabilities of ATR-FTIR. It has been further validated that serum ATR-FTIR is a valuable method for diagnosing and discriminating brain tumors from symptomatic and asymptomatic populations [80,81]. Cameron et al. [89] developed a model with a sensitivity and specificity of 94.7% and 98.4% for distinguishing meningiomas from a study involving 111 meningioma patients and 87 healthy patients. The group has even shown its potential for intracranial tumor differentiation. For example, lower-grade gliomas, glioblastomas, and brain metastases were discriminated from meningiomas with a sensitivity and specificity range of 70.9–94.4% and 81.8–86.1% [89].

Raman spectroscopy is a similar technique that also produces a unique spectral signature dependent on the complete chemical composition [90,91,92]. When a sample is introduced to a radiating laser, the non-elastic scattering effect can be measured and reflects the chemical composition of the entire sample [91,92]. Like ATR-FTIR spectroscopy, this technique utilizes the full molecular content of a sample rather than specific biomarkers.

Raman spectroscopic analysis has been established to classify healthy and meningioma individuals with high efficiency [91]. Mehta et al. [90] and Bukva et al. [91] used Raman spectra of sera or serum-originating small extracellular vesicles (sEVs) to diagnose and differentiate meningioma and glioblastoma with high specificity and sensitivity using a classification model [91]. Even so, since the sEVs also originated from other circulating vesicles, such as blood cells, platelets, and immune cells, the spectral differences could be derived from other biological responses or phenotype differences, convoluting the results [91].

A novel liquid biopsy test has further validated the success of spectroscopic approaches in cancer diagnostics. The Dxcover^®^ Brain Cancer liquid biopsy combines infrared spectroscopy with a robust diagnostic algorithm to predict intracranial disease [93]. The sensitivity and specificity of this method are comparable to other commercially available liquid biopsy techniques, and it even has the potential to discover tumors as small as 0.2 cubic centimeters [93]. The clinical utility of this test could allow for early detection and improve patient outcomes. Overall, spectroscopic techniques are a promising diagnostic tool that overcomes single biomarker limitations.

## 8. Discussion

A noninvasive liquid biopsy platform to differentiate aggressive from non-aggressive meningiomas at the time of first presentation, as well as early detection of minimal residual disease, would simplify and improve the diagnostic and therapeutic algorithms for this disease. When implemented in combination with the current imaging methods, it could allow for earlier diagnosis, improve meningioma classification, give insight into prognosis, and accurately monitor progression and treatment response.

As meningiomas are heterogeneous in their presentation and progression, liquid biopsy methods could offer the most value for differentiating meningiomas upon initial diagnosis and for tumor monitoring. Other intracranial tumors, such as solid tumor metastases or hematological malignancies, can metastasize to the brain, and the distinction between these aggressive tumors and meningiomas is limited using traditional imaging. Thus, a liquid biopsy method could differentiate meningioma from metastasis, allowing for a more rapid and accurate diagnosis and treatment.

Liquid biopsies, with the potential for complete molecular tumor characterization, could also circumvent the need for invasive tissue biopsies, particularly in cases of low-risk meningiomas in which tissue surgical intervention may not be inherently necessary. This would greatly decrease patient stress, mitigate the risks involved in intracranial operations, and continue to inform care decisions by identifying therapeutic targets and disease progression. With further improvements in precision and accuracy, liquid biopsies could replace tissue biopsies altogether, presenting all the traditional benefits of tissue biopsies with the advantages of decreased risk, more convenience, and continual monitoring of tumor progression.

This review explores many avenues for liquid biopsy in meningiomas. Potential biomarkers for meningioma include ctDNA, EVs, miRNA, seroreactivity, and spectroscopic fingerprints (Table 1). The above biomarkers have been investigated in proof-of-concept studies, but none have been validated so far, especially in a clinical setting. It is likely that different approaches may have orthogonal value in meningioma diagnostics. A liquid biopsy approach to be used in a clinical setting should be inexpensive, provide answers in an expeditious manner, and have high performance metrics to distinguish benign meningiomas from their more aggressive counterparts. Multi-omic liquid biopsy approaches will likely have the greatest performance metrics compared to a single marker approach, but it remains to be seen how accessible, expeditious, and cost-effective these approaches will be.

## Figures and Tables

**Figure 1 ijms-25-04195-f001:**
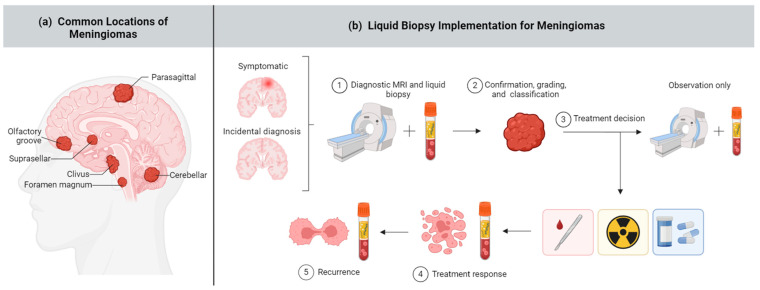
Meningioma and liquid biopsy overview. (**a**) Meningiomas can arise in several brain areas depicted above; (**b**) Liquid biopsy has potential applications across a range of meningioma diagnostic and treatment stages, including (1) early diagnosis, (2) noninvasive and comprehensive meningioma classification, (3) guiding treatment decisions including surgery, radiation, targeted therapy, or no intervention (“wait and see”) approach, (4) assessing treatment response, and (5) monitoring recurrence. Created with BioRender.com.

**Figure 2 ijms-25-04195-f002:**
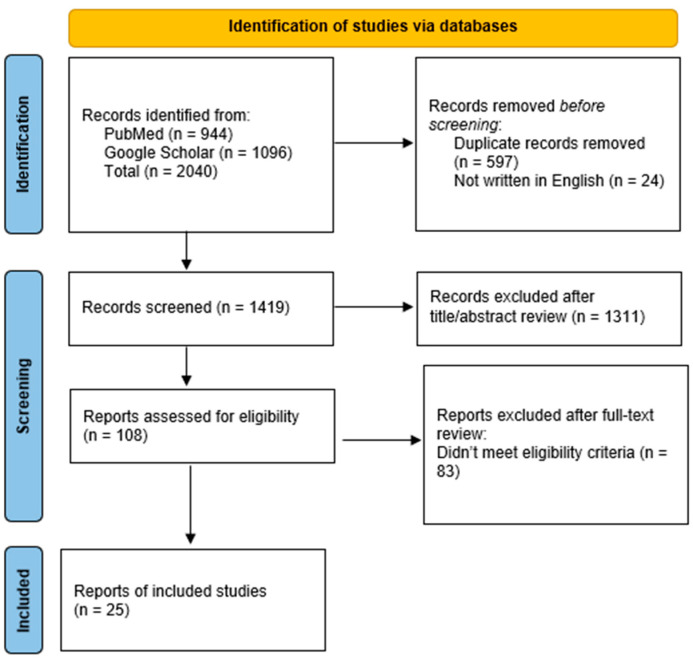
Flow diagram of studies screened for inclusion in this review.

**Table 1 ijms-25-04195-t001:** Summary of meningioma liquid biopsy studies.

Marker	Reference	Biofluid	Individuals with Meningioma	Biomarker (s) Identified	Diagnostic Models Mean Sensitivity/Specificity
cfDNA	[44]	plasma	*n* = 34	Upon gene-panel analyses of 54–73 genes, 59% of meningioma patients had at least one detectable cfDNA alteration.	N/A
	[45]	plasma and CSF	*n* = 18	Congruent mutations were detected in tumor tissue and plasma/CSF in three high grade meningiomas.	N/A
	[51]	plasma	*n* = 60	Top 300 differentially methylated regions for each primary brain tumor class.	N/A
	[52]	serum and plasma	*n* = 155	98 meningioma-specific differentially methylated probes for diagnosis and 70 risk-related differentially methylated probes for recurrence risk.	Diagnostic: sensitivity = 84.6%; specificity = 85%; Prognostic: sensitivity = 81.3%; specificity = 85.7%
EVs	[55]	plasma	*n* = 46	EV concentration is increased in meningioma patients and is correlated with extent of edema and histological malignancy grade.	N/A
	[57]	serum	*n* = 28	Matrix metalloproteinase-9 content could be a prognostic marker.	Meningioma versus healthy control: sensitivity = 64%; specificity = 61%
	[58]	serum	*n* = 61	Lower levels of miR-497 expression in exosomes of WHO grade III meningiomas compared to benign.	N/A
	[59]	serum	*n* = 74	Serum and exosomal expression of miR-497 could discriminate between meningioma and healthy patients.	N/A
	[61]	serum	*n* = 24	A group of 17 sEV proteins could discriminate between meningioma, glioblastoma, brain metastases, and control groups.	N/A
miRNA	[59]	serum	*n* = 74	miR-497 and miR-219 expression in serum could differentiate between low (WHO I and II) and high (WHO III) grade meningiomas.	N/A
	[69]	serum	*n* = 230	Six differentially expressed miRNAs: miR-106a-5p, miR-219-5p, miR-375, miR-409-3p, miR-197, and miR-224.	Meningioma versus healthy control: sensitivity = 72.3%; specificity = 81.7%
	[70]	CSF	*n* = 55	Nine differentially expressed miRNAs in CNS: let-7a, let-7b, miR-10a, miR-10b, miR-21-3p, miR-30e, miR-140, miR-196a, and miR-196b.	Meningioma versus healthy control: sensitivity = 73%; specificity = 72%
	[71]	plasma	*n* = 40	Increased expression of miR-181d is correlated with meningioma and tumor progression.	N/A
	[72]	plasma	*n* = 51	Four differently expressed miRNAs in tumor tissue and plasma: miR-21-3p, miR-34a-3p, miR-200a-3p, and miR-409-3p.	N/A
	[73]	urine	*n* = 8	23 differentially expressed miRNAs	CNS tumor diagnostic: sensitivity = 100%; specificity = 97%
Seroreactivity	[76]	serum	*n* = 24	62 antigen panel with 57 exhibiting meningioma seroreactivity	N/A
	[77]	serum	*n* = 93	57 meningioma-associated antigen panel	Meningioma versus healthy: sensitivity = 84.5%; specificity = 96.2%
	[78]	serum	*n* = 53	1827 antigen-expressing clones	Meningioma versus healthy: sensitivity = 91.8%; specificity = 95.6%
	[79]	serum	*n* = 24	Five immunogenic epitopes of TXNDC16 could discriminate meningioma and control serum.	Meningioma versus healthy: sensitivity = 90%; specificity = 83.7%
Spectral Output	[80]	serum	*n* = 5	ATR-FTIR spectral output	CNS tumor diagnostic model: sensitivity = 81%; specificity = 80%
	[81]	serum	*n* = 46	ATR-FTIR spectral output	Intracranial tumor diagnostic: sensitivity = 93.2%; specificity = 92.8%
	[88]	serum	*n* = 47	ATR-FTIR spectral output	Meningioma versus glioma: sensitivity = 66.7%; specificity = 82.1%
	[89]	serum	*n* = 111	ATR-FTIR spectral output	Meningioma versus healthy control: sensitivity = 94.7%; specificity = 98.4%
	[90]	serum	*n* = 35	Raman spectral output	N/A
	[91]	serum	*n* = 28	Raman spectral output of serum-derived sEVs	Control vs. meningioma: sensitivity = 80%; specificity = 85%

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
