# Peer review of "Innovation in Non-Invasive Diagnosis and Disease Monitoring for Meningiomas"

_ijms, 2024, doi:10.3390/ijms25084195_

Round 1

Reviewer 1 Report

Comments and Suggestions for Authors

This paper is a review about liquid biopsy in meningiomas diagnosis and follow-up. 

Liquid biopsies is a promising research field in many tumors, however this review does not add data or reflection to already existing literature. In addition, it lacks precision, both in the methods (not a systematic review), and in the terms employed in general (using "brain cancer" in a paper about meningiomas sounds very vague). 

We suggest that the authors rewrite the paper as a shorter review if they wish to share their experience, and focus on some main messages, rather than propose an overview that lacks convincing information. Especially, they need to focus on the benefit liquid biopsies could bring to the patients care, which seems unclear to a neurosurgeon following hundreds of patients with benign meningiomas without need for additional expensive molecular testing. They could also share their expertise on the most promising technique in this specific research area. 

Comments on the Quality of English Language

none

Reviewer 2 Report

Comments and Suggestions for Authors

Dear authors, 

First of all, I’d like to give a great congratulation to them for nice and graceful review. Korte et al. did nice review of liquid biopsy methods which exploit circulating tumor biomarkers such as DNA, extracellular vesicles, micro-RNA, proteins, and more, offer a non-invasive and dynamic approach for tumor classification, prognostication, and evaluating treatment response mainly focused on intracranial meningioma. Up to dates, it became hot topic for the diagnosis and predicting prognosis of cancer. They provided the extensive and comprehensive information of updated research of the liquid biopsy for intracranial meningioma.

However, to be honest, it is difficult to predict whether these studies can be helpful in terms of clinical application. Because meningioma is a typical benign brain tumor, it is a disease with standardized evidence for treatment or diagnosis. Furthermore, unlike other solid cancer, meningioma is a tumor limited to the intracranial cavity and is not a disease that invades the whole body without metastasis to the other site of body. The assumption that diseases limited to the intracranial cavity are evaluated with biological indicators affecting the whole body, such as serum, is not realistic in itself. Of course, the process and procedure of the study have been scientifically conducted, but it is unlikely to be helpful in clinical settings using the results. Therefore, if this study is limited to malignant meningioma, not benign meningioma, the results are expected to be helpful in the actual clinical field. Good luck.
